# “I Don’t Want to Be Known as a Weak Man”: Insights and Rationalizations by Male Students on Men’s Sexual Violence Perpetration against Female Students on Campus

**DOI:** 10.3390/ijerph20054550

**Published:** 2023-03-03

**Authors:** Yandisa Sikweyiya, Mercilene Machisa, Pinky Mahlangu, Ncediswa Nunze, Elizabeth Dartnall, Managa Pillay, Rachel Jewkes

**Affiliations:** 1Gender and Health Research Unit, South African Medical Research Council, Pretoria 0001, South Africa; 2School of Public Health, Faculty of Health Sciences, University of the Witwatersrand, Johannesburg 2193, South Africa; 3Sexual Violence Research Initiative, 28 High Street, Pretoria 1045, South Africa; 4Office of the Deputy Director General, Care and Support Services, Department of Basic Education, 222 Struben Street, Pretoria 0001, South Africa; 5Office of the Executive Scientist, South African Medical Research Council, Pretoria 0001, South Africa

**Keywords:** men, sexual violence, sexual harassment, masculinities, college, South Africa

## Abstract

Understanding how men view rape is foundational for rape prevention, but it is not always possible to interview men who rape, especially in a college campus context. We explore male students’ insights into and rationalizations for why men on campus perpetrate sexual violence (SV) against female students by analysing qualitative focus group discussion data with male students. Men contended that SV is a demonstration of men’s power over women, yet they did not perceive sexual harassment of female students as serious enough to constitute SV and appeared to be tolerant of it. Men perceived “sex for grades” as exploitative and rooted in the power asymmetry between privileged male lecturers and vulnerable female students. They were disdainful of non-partner rape, describing it as acts exclusively perpetrated by men from outside campus. Most men felt entitled to have sex with their girlfriends, although an alternative discourse challenged both this entitlement and the dominant masculinity linked to it. Gender-transformative work with male students is needed to support them to think and do things differently while they are on campus.

## 1. Introduction

Male-perpetrated rape is a global phenomenon that has serious impacts on women, children, and men who experience it [1,2,3,4,5,6]. Research with men in the general population in South Africa has pointed to key risk factors for rape perpetration [4]. These include men’s experience of sexual violence (SV) and neglect in childhood and exposure to violence in the home as a child, especially witnessing the abuse of their mother, as well as a personal experience of trauma as a child or adult (having been raped themselves). Moreover, men who perpetrate non-partner rape are much more likely to engage in a range of other violent and antisocial behaviours including having been in gangs, using drugs, fighting with other men, and having weapons [4,7].

Qualitative research from South Africa has provided critical insights into the local social, economic, and political conditions that shape men’s use of violence, including rape against women and girls. These include the historical and continued political and economic marginalization of, and the high unemployment rate among, black African men [8,9] deprivation, emotional neglect, disintegrated family lives, and construction of violent masculinities [10,11,12]. 

Another body of work has shed light on the multiple-perpetrator rape or gang rape phenomenon [13]. Through her ethnographic research with young people in Mthatha, Wood [14] described four group-rape scenarios in which young women were raped by men who were either strangers, acquaintances, suitors, or boyfriends. The first scenario was described as “an opportunistic and usually alcohol-motivated group rape often perpetrated by stranger men”. The second scenario occurred in circumstances where a group of friends took sexual advantage of girls with whom they had been drinking, who were drunk to the point of being unable to resist. The third scenario typically happened where a girl who had refused one of the group’s sexual advances was punished by being raped by the man who propositioned her and by one or more of his friends. The fourth scenario typically occurred when a young man organized for his friends to have sex with his girlfriend as a way of ending the relationship when he was tired of her [14]. 

Globally and in South Africa, studies on men’s rape perpetration have largely been conducted among working-class and economically marginalized men. As such, Phipps [15] argues that “working-class men, unemployed and marginalized black African men are constructed as more likely to commit sex crimes than their middle-class counterparts and are more often prosecuted and convicted, as well as receive harsher sentences upon conviction” (p. 677). Yet, middle-class and white men who commit SV and rape against women are more likely to use their financial power and privilege to evade arrest, prosecution, and conviction [15]. Thus, the literature on men’s rape perpetration must not be read as claiming that more marginalized men and black African men are more prone to committing SV.

Within the sub-Saharan region, the starting point for research to understand men’s perpetration of rape of women is a theoretical understanding that rape of women and girls is driven by patriarchy [3,16] and specific understandings of masculinities [17]. The key theory that has informed research in the sub-Saharan region on men’s perpetration of violence, including rape, has been Connell’s theory of gender and power [18]. The critical scholarship on men and masculinities has primarily employed Connell’s concept of hegemonic masculinity to elucidate the rationale and means by which gender inequality is preserved. Concerning hegemonic masculinity, Connell contended that in any given context, there are several masculinities; however, among these exists a shared cultural model of masculinity that is seen as ideal and superior to the other existing masculinities [19,20]. 

Gender scholars have highlighted the importance of gender-transformative interventions to shift harmful gender and social norms, and gender-inequitable attitudes amongst men, and support them to construct nonviolent masculinities [21,22]. Researchers and programmers in low- and middle-income countries have implemented several programmes to engage boys and men in transforming masculinities [16,23]. Interventions working on transforming masculinities generally aim to support critical reflections about masculinity and violence and challenge the acceptability of violence [16,17,24]. Some of these interventions also address recognized risk factors for SV perpetration [25,26]. 

In the Global North, college samples are overrepresented in the SV literature base [27], yet this contrasts sharply with the limited literature on this topic from the Global South. While numbering few, studies in South Africa have highlighted the need to develop gender-transformative interventions for use with men in higher-education institutions (HEIs) [28], and the first step in developing these interventions is to conduct formative qualitative studies to gain an appreciation of men’s reasonings and rationalizations for perpetrating SV on female students, thereby advancing our knowledge on the context-specific risk factors for male-perpetrated SV in HEIs. The current study aims to address this gap. This study aims to describe the context and circumstances under which SV experienced by female students occurs in selected HEIs. Moreover, the study seeks to gather insights of male students on what they thought renders female students vulnerable to SV on or off campus. 

### Context of the Study

This qualitative study was conducted as part of formative research that aimed to inform the development and pilot test of Ntombi Vimbela!, a sexual- and gender-based violence (SGBV) prevention intervention for use with female students (18–30 years) in HEIs in South Africa. Further descriptions of the Ntombi Vimbela! intervention development and pilot study are provided elsewhere [29,30]. As part of conceptualizing and developing male students’ component of the Ntombi Vimbela! intervention, a need was identified to first understand the contextual factors underlying SGBV perpetration by men in South African HEIs. Specifically, we wanted to enhance our understanding of the influence of the context/environment in Technical and Vocational Training colleges (TVETs) and universities in predisposing students to SV experiences (females) and perpetration (males) to help develop a theory of change and logical framework for the intervention, and inform the adaptation of interventions that have been found effective in other settings to the South African context. Thus, through this qualitative study, we sought to gather men’s views on factors that influence female students’ risk of experiencing GBV on or off campus and increase men’s sexual aggression and perpetration of SV against female students. 

## 2. Materials and Methods

This was a qualitative study conducted with volunteer male students (aged 18–30) who were purposively selected from some TVETs and universities in South Africa. Registered male students who had been on campus for at least a year were eligible to participate as they were deemed to be familiar with the campus life and dynamics and, thus, could provide insights on the topics explored in this study. 

We conducted one focus group discussion (FGD) in each of the eight campuses—i.e., six TVET campuses and two university campuses in 2018–2019. All the study sites were in historically disadvantaged institutions (HDIs) and were in five provinces in South Africa: in the Eastern Cape, KwaZulu–Natal, Limpopo, Gauteng, and Mpumalanga. The study sites were selected with the input of the National Department of Higher Education and Training (DHET) and management committees of the selected institutions and were situated in urban and rural locations, therefore catering to diverse student populations.

We conducted eight FGDs with 88 male students. Almost all the participants came from resource-limited urban townships and rural villages, with many of them reporting that they came from disadvantaged family backgrounds. Owing to this, their studies were funded by the South African government’s National Student Financial Aids Scheme. Each FGD comprised between 10 and 12 students, ran for 1–1.5 h, and was facilitated by the first author, who is an experienced male qualitative researcher. Within the campuses, the locations at which the FGDs were conducted were carefully selected to ensure that no one outside the room was able to hear the discussions. A guide with open-ended questions was used to facilitate the discussion. In the FGDs, men were first asked how common they thought SV against female students was on their campus, who were the likely perpetrators, and to describe the characteristics of men who perpetrated it. Men were further asked to share their opinions of what predisposes female students to SV victimization, and what makes men on their campuses perpetrate SV against female students. 

Before the start of the FGDs, it was emphasized to the participants that they were being asked to talk generally about SV incidents against female students, and not about themselves. It was explained to them that they were being asked to share stories of SV against female students that they had observed on campus, without mentioning the names of the perpetrators. 

Moreover, before the start of each FGD, the facilitator emphasized to the participants that the FGD was not an opportunity for them to talk about how to trick women better or glorify their sexual exploits. The FGDs were conducted mainly in English, although some participants expressed themselves in vernacular. All participants gave written consent to participate and for the FGDs to be recorded. 

We analysed the data inductively using a latent content analysis approach. However, there were deductive elements to the analysis, as we explored themes that were related to sexual harassment and the coercion of female students. The first author (YS) and co-authors (MM, PM, and NN) were involved in coding the transcripts. The other co-authors (ED, MP, and RJ) were involved in the later stages of data analysis and interpretation of the findings. First, we read the transcripts repeatedly to familiarize ourselves with the content of the transcripts. Afterward, we established broad codes which comparatively resembled the questions in the FGD guide. Subsequently, the text which seemed to fit together was clustered under a specific code. Additionally to this, we explored the data and identified numerous open codes. Analogous open codes were then grouped under clearly defined themes. In cases where there were divergences in the codes, we settled them by verifying the codes against the data. As the next stage of the analysis, we explored the relationships between the themes and interpreted what we saw emerging. 

Ethics approval was obtained from the South African Medical Research Council’s Human Research Ethics Committee. Permission to conduct the research was obtained from the Department of Higher Education and all participating higher-education institutions. Participants provided voluntary, written informed consent, and were reimbursed with ZAR50 for their participation. Refreshments were provided during FGDs. 

## 3. Results

Our data pointed to various circumstances in which SV was perpetrated in South African higher-education institutions. First, some men spoke of acts of sexual harassment which were rooted in men’s feelings of inadequacy and stemmed from attempts to get a female student’s attention. Second, examples of abuse of power by men on campuses in contexts of sex demanded for academic favours or assistance given with accommodation or financial challenges. Third, descriptions were given of rape incidents where men alone, or in groups, would go out to find a young female student whom they could rob of her possessions and rape. Fourth, some men provided descriptions of acts of rape of female students by peers (and possibly by themselves). When the men spoke of the three scenarios occurring in the college community, they suggested there was tolerance and normalization of the first one, but they were highly disdainful of the second and third. When discussing the stranger rape scenario, they perceived this in a way that was entirely different and went so far as to say it was not something that any man from campus would do. They perceived a complete separation of this form of rape from other acts of sexual harassment, sexual coercion, and student-perpetrated rape in college dorms and parties.

### 3.1. Spectrum from Sexual Harassment to Rape

Some participants gave examples of how sexual harassment of female students by male students occurs: 

Is it because I am not fit for her? or whatever and then I get to do things that will constantly harass her (a female student), when I see her I grab her forcefully, maybe I try to force a kiss, or I spank her buttocks or things like that. 

The unspoken questions that ”is it because I am not fit for her?” or “I’m not good enough for her to notice me?” suggest that men saw these acts as attention-seeking and did not perceive themselves as having and using power over women, even if they used sexual power in the process of harassment. Moreover, feeling inadequate, some men spoke of using drugs to muster the courage to take the harassment a stage further to rape: 

The other thing is that the types of drugs that people use are different. Some make people very violent. So, they differ, there are those that after you had taken them you feel very powerful and want people to see that, as a result, they go to a girl that they have always wanted, knowing that if that girl refuses, they will take them to the bushes by force and do whatever they want with her because he is acting under the influence of drugs.

### 3.2. Sex for Favours

Participants also described “sex for grades” as a particularly common phenomenon, with it mostly occurring when either a male lecturer offered to improve a female student’s grade in exchange for sex or female students approached a male lecturer to query low or unfavourable assessment outcomes. This was described as presenting opportunities for some male lecturers to take advantage of desperate female students and demand sex in return for improving their grades or remedial academic support: 

If Mr X. [lecturer] is doing favors for you as a [female] student, like if you were absent on the day a test was written when you return the following day, you cry to Mr. X. [but] you don’t have a sick note, you don’t have anything, and he allows you to write. The next thing after a few days, he can come back to you and say, “since I did you a favor, I also in exchange want a [sexual] favor”. 

It was not only “sex for grades” that men said was used to coerce sex with female students, but also the provision of material benefits. Some participants said some young male lecturers would “befriend beautiful and attractive female students, give them money, gifts or other material things, and later demand sexual favors from them”.

Other participants shared accounts of male lecturers and Student Representative Council (SRC) members who were known for providing financial support to female students and thereafter demand sexual favours from them. Such men were said to be “powerful” or “influential” on campuses and to take advantage of the “vulnerability of 1st-year female students”, as well as other female students, and coerce them into sex through promising them admission at the institution or finding them accommodation in exchange for sexual favours:

At the beginning of the year, campaigns are run to help new students get admitted, so these people who are doing those campaigns have accommodation on campus, so a new student comes from home and knows nothing about the institution, so other people from those campaigns take advantage and promise to give the girls accommodation and then after they had given them a place to sleep they expect to get something [sex] in return or maybe he says ‘no I will make ways for you to get admitted so you and I will have an agreement’ and that will lead to sexual violence.

Many participants perceived 1st-year female students to be “naïve”, and “confused” during their first few months of arrival on campus, and often lacking money, accommodation, and other necessities (e.g., food and clothing), which made these female students “vulnerable”. One participant explained that: “another thing that we can take a look at, at the beginning of the year some girls are still confused, and they don’t have accommodation so it’s another chance for them to get raped”. 

When asked whether the National Student Financial Aid Scheme (NSFAS) bursary (i.e., government grant provided to students in financial need) mitigates this risk, participants said NSFAS is usually paid late into individual students’ accounts, especially to first-year students, enabling some senior male students who had already received their stipends to offer money to needy and vulnerable new female students and later demand sexual favours: 

You will notice that the NSFAS process takes a long time, those who have been here for years quickly get funded by NSFAS, and first years get funded later in the year, then those who got funded will go to these [new female] students to give them something maybe R300 or R500 knowingly that after that he will get something [sex].

### 3.3. Stranger Non-Partner Rape

Non-partner rape incidents were described as largely opportunistic acts which occurred in conjunction with other criminal activities such as break-ins, robberies, or cellphone thefts. The participants spoke of non-partner rape of female students as acts exclusively perpetrated by men from outside campus: 

So, several rapes that are occurring outside campus, is because of the [men from the] community. You can’t find a guy on campus going outside to say. “I’m going to rape someone outside”, that’s rare … but for guys who are staying outside as community members, they can go out and say “we are going to … break into [her rented room] and rape the lady”.

One participant gave an account of the gang rape of a female student by men from a neighbouring community: 

There is a story that happened last week, they raped, [it was] a group of men, they took the girl to the bushes and raped her many times and then ran away but some of them were found. And then this week there’s that [court] case here in town, some are still on the run, but some are in jail … She is a student here. But the situation happened outside. 

### 3.4. Male Students’ Sexual Coercion and Rape

Despite a prominent discourse of rape being perpetrated by non-students, there was considerable space given in the FGDs to describing acts by their peers (and possibly themselves) of rape of female students. 

Some participants spoke of men within their campus who attended parties or bashes with a view that, when drunk, female students are “easy targets” as they are not able to communicate their refusal to engage in sex [14]. One participant explained that: “once she is high the guys just know that they will have sex with her today because she is asleep [blackout from alcohol], it’s easy for the guy to just jump on top of her and have sex with her”. Other participants spoke of men who spiked the drinks of female students so that they black out, making it easy for the men to have sex with them:

So, for me to get what I want, I come up with another strategy. Some [men] put something in a drink. I know she will be knocked off, and then we go together. The moment she wakes up in the morning, she realises she is naked, and we are in bed. That’s the first thing when she wakes up, she would be like “what happened”, and you say “don’t you remember? We had sex and you agreed, you are the one who kissed me”, she doesn’t even have a clue what happened. That’s why she doesn’t even go and report it, but actually, that was rape in some way. 

In this extract, although he speaks of “for me” in the context of the FGD, it was clear he was not necessarily talking about himself. 

Some of the narratives from some participants suggest that incidents of gang rape of female students are chiefly carefully planned and executed: 

The thing is when we are going to a party, as I’ve said we agree that “you know what, I’m going to sleep with this girl”. We pinpoint like that “I’m going to go with this one”, and “if you don’t mind, I’m going to offer you this one”. So, when we have a party at the lodge, you know six guys who are staying there, each and everyone is going to sleep with that girl. So, they don’t care whether that lady blackouts in the room or what, they just want to sleep with her.

Most participants said men are socialized to be in competition with each other in their conquests for women and that they discuss pursuing women within their peer groups. This competition appears to be driven by both the need to “impress male friends” and avoid being seen “as weak”, and this, in turn, made some men “try by all means” to sleep with a girl, even if this meant sleeping with her by “force”: 

Another reason for us to force ourselves on females, … I want to impress my friend, let’s say me and my friend we have met these girls at the same time and maybe his girlfriend is sexually active so he immediately sleeps with her then he will come and brag to me, mind you I haven’t slept with my girlfriend yet and I don’t want to be known as a weak man as that will make me feel low. So, I will try by all means to sleep with this girl then I go and tell my friend that I finally slept with her, so I didn’t want my friends to say I am weak, so I decided to force myself on her. 

Indeed, some participants discussed how they resolve the need to be seen as powerful and “not be labelled as a weak man” by their male peers by coercing sex with female students, as seen in the extract below: 

Another thing we [male students] don’t want to spend two weeks without sleeping with a girl. When you meet a girl, you want to sleep with her immediately so that you won’t be labeled as a weak man, because my friends would ask, have you slept with her and when I say no they will laugh at me, so that is another reason why we are forcing ourselves on girls.

Many other participants said when female students “showed their bodies too much”, to prove their manliness, they had to have sex with them, including through “force”, to avoid being thought of as “weak” by their peers: 

But they show their bodies too much as a result, even if you were not interested in her you realise that if I can keep quiet others will think that I am weak, so those are the things that drive us [to force ourselves on them] because they are naked.

Also alluding to men’s avoidance of being perceived as weak, one participant explained that when a man is told that he is weak, that is a demonstration of “disrespect” and being perceived as less of a man:

Let us start this way, one, firstly, when you are told that you are weak, you are being disrespected, it’s a big humiliation and it’s like you are worthless, other men will view you like you are not a man enough.

While some participants argued that “you can’t find a guy on campus going outside [campus] saying. “I’m going to rape someone outside””, thus contending that non-partner rape is “not something a man from campus would do”, the extracts above clearly describe rape. In so doing, there was some level of awareness that men recognized that their actions constituted rape. However, at the same time, as it is perpetrated by students on campus, many of the men viewed it as different, and if she did not report it, they considered it “not rape”.

Sentiments of male students’ sexual entitlement within intimate relationships were also shared, with some participants indicating that they do not perceive that their girlfriend had a right to say no to sex when he wants to have sex with her. They might start by coercing her into sex through manipulation saying: “Yes if you say you love me prove it by having sex with me”. However, if that did not work, men described forcing sex. Indeed, many men equated having a girlfriend with being given complete sexual access to the girlfriend, and this was clearly expressed in one participant’s assertion that “forcing a girlfriend to have sex” with him is not rape as “she is your girlfriend”. In his own words: 

She is your girlfriend, and she agrees to visit you in your room but when she is there now, she refuses to have sex with you. Then you ask yourself why she agreed to visit you if she does not want to have sex with you? She knows when you take her to your room as a guy you want sex, so in such cases, you cannot control yourself, so you end up forcing her into having sex with you and then people say it is rape, yet she is your girlfriend. 

### 3.5. “The Dad Is Not the Head”: Respectful and Gender-Equitable Men

While our data have foregrounded endorsement of gender-inequitable attitudes, norms, and practices which are seen through most participants’ tolerance and acceptance of sexual harassment and rape of female students, there were several instances where alternative discourses emerged in the FGDs. These discourses chiefly challenged the narratives that sought to position men as “naturally superior and dominant to women”. Some participants expressed disapproval of gender-inequitable attitudes and norms, and denounced rape myths, with their narratives suggesting that they ascribed to alternative masculinities. 

These participants who held equitable gender attitudes, while few, strongly expressed support for gender equality. Arguing for gender equality between women and men, one participant said: “there is no one above the other in the house, the dad is not the head, he was the head before but now he is not”. 

Emphasizing the significance of respect in intimate relationships, other participants described how they treated their intimate partners with respect and as equals: 

A relationship is a part of being in love, and respecting each other … let me say it is a connection, a good connection on how we relate …, it is part of the relationship that I respect the girl that I am dating and treat her in a right way, you understand? 

Some participants boldly challenged their peers’ endorsement of rape myths, particularly around the notion that women who wear miniskirts or revealing clothes are challenging men to pursue them for sex and thus cannot complain when they have been raped: 

The [female students’] dress code is what we [men] are complaining about, right? At the end of the day, it’s a free country anyone can dress however they want. I think that we as men take it too far…, if things were in reverse and it’s your sister who is walking over there, you wouldn’t comment even if she is wearing leggings, you’d keep quiet because of the respect you have for your sibling.

Another participant argued that how a woman dresses does not signal a desire to have sex and that accepting money, drinks, or gifts from a man does not mean a woman has to agree to have sex with the man: 

Personally, I don’t think… if a female wants to be fashionable, if she wants money, she wants to go to Cubana (a club), personally, I don’t think that’s a problem. Because the guys are the ones who say yes to all of these things. So, at the end of the night, if she doesn’t want to [have sex with you], she doesn’t want to. Just because you went out with her, that’s irrelevant. And another thing, the way they [women] dress, that’s irrelevant also. When they wear shorts when they wear skirts like it’s irrelevant. Because I’ve noticed, people get raped, like infants, so clothes for me personally, they don’t mean anything. 

## 4. Discussion

Our findings provide useful insights into contexts in HEIs that create vulnerability to SV victimization for female students and perpetration by men. Specifically, the findings give us critical insights into the workings of rape culture on campus, male students’ sexual entitlement, and our participants’ thinking around SV against female students. As there is a dearth of studies on college men’s perpetration of SV in HEIs in South Africa, these insights are critical, as they highlight areas to target in SV prevention interventions in HEIs in South Africa [28].

This study suggests that a combination of factors may be informing men’s use of SV against female students. First, our findings suggest that when it comes to men’s perpetration of sexual coercion and rape, men knew they could exercise power over women and get away with their violent actions with no consequences to them [14]. These findings are akin to those of other studies in patriarchal societies in sub-Saharan Africa [31]. The study findings also show that a few participants in our sample demonstrated gender-equitable masculinity by rejecting patriarchal gender norms and attitudes and providing accounts of gender equality in their intimate relationships. 

In line with the existing literature [32,33,34], our findings suggest that male students in HEIs are under intense peer pressure to prove their virility, masculinity, and sexual prowess to male peers. Indeed, some participants reflected on how the pressure to have sex with new girlfriends to prove their manhood to peers made some male students rape their new girlfriends. Yet, while some acknowledged that their actions constituted rape, many considered forcing women, especially their girlfriends, into sex to be acceptable and normal [14]. This finding suggests a perpetuation of “rape culture” [35] by men in our sample and reflects other studies which have shown that rape culture is widespread on South African campuses [36] and in society [37], and further demonstrates how peer pressure and conformity to hostile masculine norms operate to make men on campus perpetrate SV against female students [34]. This finding on the existence of rape culture on South African campuses suggests that it is important to engage male students in gender-transformative work and teach them to do things differently while they are on campus [22].

Additionally, the study findings reveal how male students in HEIs engage in competition and sexual conquest for female students and discuss this within their peer groups as a way to bond as men, while simultaneously dehumanizing female students to rationalize their SV on them. Scholars have described fraternal organizations, networks, and house parties as spaces within campuses where male students support and celebrate male peers who sexually objectify and disrespect female students [36,38,39]. These findings suggest the construction of a hegemonic student masculinity in South African HEIs that is very sexually violent and characterized by sexual conquest, control, and dominance over female students [36]. Gender-transformative interventions in HEIs must create safe spaces for male students to critically reflect on how the dominant male youth culture on campuses encourages and reinforces men’s SV behaviour towards female students. We contend that challenging men’s engagement in competition with each other, and their use of demeaning language towards women, could be important entry points for shifting their gender-inequitable attitudes and violent masculinities. 

This study suggests that the sexual harassment of female students in HEIs is not only considered common but is also acceptable to and justifiable by most of our participants, and not considered serious enough to constitute SV. Phungula [40] contends that “once sexual harassment practices occur repeatedly, overlooked, disregarded and unchallenged, this is likely to send a message to both male and women students that women can be treated with a lack of respect and also violently as it does not matter to anyone” (p. 10). To shift this mindset, programmes that address the topics of sexual harassment and sexual consent are needed in South African HEIs to enable both male and female students in these contexts to understand what these are, and for men to understand when their language and actions are harmful and coercive. Furthermore, there needs to be stricter enforcement of sexual harassment policies in HEIs and ensure the implementation of punitive actions for perpetrators. 

Our findings suggest that some men think of young first-year female students as naïve and vulnerable, making them believe that these students are easy targets for coercing into sex. HEIs, especially those who cater to students from disadvantaged backgrounds, need to improve their systems to ensure that their student admission processes and financial aid reduce the vulnerability of first-year female students during their early days on campus. This should include screening men who volunteer to be involved in orientation processes to assess their gender attitudes, with only those demonstrating equitable gender attitudes selected to be involved in the orientation of first-year students, with their work closely supervised. 

Some men in our sample espoused gender-equitable masculinities, displaying this through advocating for gender equality and being respectful to women, including intimate partners. This finding is important and should be used as a window of opportunity for engaging men in HEIs in gender-transformative work. Research has shown that having equitable gender attitudes decreases a man’s likelihood of perpetrating IPV [41]. Our findings appear to indicate that, through their assertions, practices, and beliefs, these men rejected the hegemonic masculinity ascribed to by their peers. However, this was not necessarily without cost to them, as they reported that they were considered weak by their male peers. Notwithstanding, the fact that these men publicly challenged hegemonic masculinity, inequitable gender attitudes, and norms and rape myths in the presence of their peers who supported these suggests that they may be amenable to gender-transformative work which seeks to prevent men’s perpetration of SV and promote respect for intimate partners and gender-equitable interactions with women. As part of gender-transformative work, male students who ascribe to gender-equitable masculinities should be identified and supported to be advocates for gender equality on their campuses and drive positive messages regarding gender equity and respect for women and anti-violence. Indeed, studies have shown that gender-transformative programming is effective in supporting men to shift harmful gender norms and construct transformative masculinities [28]. The Stepping Stones [42] and Stepping Stones and Creating Futures interventions [23] are some of the gender-transformative programmes developed and evaluated in the Global South that have been shown to reduce men’s use of violence and the construction of violent masculinities. 

### 4.1. Implications for Interventions and Research

In our study, the participants were split between those who spoke as men who rape or sexually violate female students and those who created a distance from such men by presenting gender-equitable views. Existing interventions tend to not target the second group, but more so the first group. As such, there may be a benefit from the incorporation of bystander components in extant interventions to target men who claim not to engage in SV themselves [43]. The interventions working with the first group should focus on self-esteem building—engaging with perceived ego fragility, engaging with constructions of masculinity and challenging sexual entitlement, discussing consequences of rape, and supporting them to build their empathy and understanding of women and their experiences of SV. Additionally, there is a need to implement structural approaches to preventing SV against female students perpetrated by men in positions of authority within HEIs. Institutions should consider implementing computer-based assessments which record the time of tests or exams so that the option of bribery for a late test is shut down, or having two markers for papers, and beefing up information for students to avoid female students falling foul of predators over-accommodation and grants. Furthermore, there is a need to work with female students to challenge the idea that they “owe” anyone sex. 

To take forward the critical, but limited, research on SV perpetration in South African HEIs and advance knowledge on gender-transformative programming with men in HEI settings, there is a need for campus-wide violence perpetration surveys to identify perpetrators of SV against female students. Researchers could thereafter design follow-up qualitative studies, using one-on-one in-depth interviews, with both men who had reported acts of SV perpetration against women in the past 12 months and those who had not, to explore in-depth the context-specific men’s motives and rationalization for perpetrating such violence against female students. Including both the perpetrators and non-perpetrators will mask both the interests of the study and prevent the participants from being identified as SV perpetrators on campus. Furthermore, the qualitative findings presented in this paper can inform the development of future SV prevention interventions, targeting men in the Global South. These interventions could then be evaluated through prospective cohort studies.

### 4.2. Reflexive Discussion on Ethics of Listening to Men’s FGDs on Sexual Violence Perpetration 

In terms of South African laws, the researchers had no legal obligation to report the disclosures of rape described by the participants in this study, as none of the disclosures reported in the FGDs appeared to have been committed against minors. 

Additionally, the violence and rape disclosures in the FGDs were done in such a way that the respondents shared no specific information that would have enabled the identification of the person who was violated. Moreover, the actual names of the perpetrator (e.g., name of faculty member) were not revealed, and most cases of SV discussed were described as having been reported to the authorities including the institution management and police. Additionally, to avoid disclosures that would have made the researchers perceive an ethical obligation to report to authorities, the researchers requested the participants not to mention any person they were referring to by name or describe them in such a way that the person could be identified. 

Notwithstanding, the FGDs contained degrading comments about women and trivialization of men’s violent practices towards women. While it is essential in studies on men’s perpetration of SV against women to allow men to talk candidly about their gender attitudes and SV practices, as that enables researchers to better know what men think and how they rationalize their use of SV against women, it is equally important to use the FGDs as an entry point and space to challenge men on their gender-inequitable attitudes and SV practices towards women. In our study, during or immediately after the conclusion of the FGDs, the first author engaged the participants in conversation regarding their gender-inequitable attitudes, as well as their support and trivialization of SV practices. He pointed out that some of their beliefs and gender-inequitable attitudes and violent practices towards women were harmful to women. In those engagements, he challenged the men to critically reflect on the consequences of their gender-inequitable attitudes and men’s violent practices towards women, women’s health and lives, and themselves. This elicited healthy and vibrant deliberations between the first author and the participants, with some men indicating that the FGDs were a rare opportunity for them to debate gender-related issues with their male peers and reflect on their behaviour. Several men indicated that their socialization as boys had shaped their gender-inequitable attitudes toward women. While we do not claim that these engagements resulted in shifts in the men’s gender-inequitable attitudes, several men said they appreciated the “new information and way of thinking” about gender and gender relations that they obtained during the FGDs. Moreover, many men expressed that they wished that the FGDs, on the topics explored this study, were not a one-off event on their campuses. This may suggest these college men’s inclination to be engaged in gender-transformative programming. 

### 4.3. Limitations

Our study sample comprised men who had been students in the selected institutions for more than one year; it was a small sample as we only had one FGD per institution. This criterion was vital to enable us to talk to men with more experience in college life, but limits what we can say about any one institution. Our study was qualitative; thus, our findings are not generalizable. However, we hope that our findings will be useful in generating hypotheses for future SV perpetration research in HEIs and advance knowledge on methodologies for studying men’s SV perpetration in HEIs.

Our study did not aim to focus on individual men’s gender attitudes and violent sexual behaviour, which could have been achieved through in-depth interviews with male participants who identified as perpetrators of SV. As qualitative methodology is largely dependent on self-reporting, it tends to be biased by internal states only known to the informant and by their decision to share (or not share) them with researchers [44]. As noted by DeKeseredy and Schwartz [45], college settings are settings with many likely SV perpetrators, yet very few people tend to be reported as having offended or having sexual assault criminal records. We concur with DeKeseredy and Schwartz [45], and further argue that, in the current study, the feasibility of recruiting known SV perpetrators in the campus settings was near-impossible, given that such conduct warrants expulsion. 

We acknowledge that during the FGDs, the participants’ perceived male peer norms may have resulted in problematic silence, group thinking, and social desirability bias. Some participants may have felt obliged to respond in a particular manner in the FGDs, including feeling the need to perform the idealized masculinity and represent themselves positively in the presence of their peers. Additionally, it is possible that some men sought to represent themselves in a socially desirable manner concerning their gender attitudes and behaviours. Moreover, while our participants openly shared their insights and rationalizations in the FGDs, it is important to highlight that people are not perfect at appreciating or explicating their own motivations for how they acted or behaved. With regards to SV perpetration, the participants shared their perceptions and did not necessarily identify themselves as SV perpetrators, even though some of their accounts may have suggested so. Additionally, it is difficult to definitively know whether the participants were speaking about themselves or other men when describing violent sexual behaviours against female students on their campuses. Furthermore, it is not unusual for African people who are not English-first-language speakers to speak about issues or incidents as though they are referring to themselves or were involved in them, as a way of describing a general phenomenon or public account.

## 5. Conclusions

Our findings suggest the need to engage men in HEIs in shifting their thinking around how they perceive female students, challenge their views around sexual entitlement and sexual consent, and support them to learn to treat women with respect. However, a theme that was completely absent from the narratives of men in our study was that of consequences for SV—it is thus important that in gender-transformative interventions in HEIs, spaces must be created for men to reflect on the consequences of SV for women who are victimized and men who perpetrate it. 

## Data Availability

The data presented in this study are available in the article.

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
