# Peer review of "“I Don’t Want to Be Known as a Weak Man”: Insights and Rationalizations by Male Students on Men’s Sexual Violence Perpetration against Female Students on Campus"

_ijerph, 2023, doi:10.3390/ijerph20054550_

Round 1
Reviewer 1 Report
Goal: This study aims to elicit motivation and common scenarios of perpetration of sexual violence and harassment of women from men enrolled in higher educational institutions (HEIs) in several locations around South Africa. This study is part of a larger project to develop a GBV/SV prevention intervention for use in HEIs in South Africa.
This is both useful research and a well written manuscript. My comments below are meant to expand and clarify, rather than correct what is already a solid contribution to the literature.
The Abstract opens with the important assertion: “Understanding how men view rape is foundational for rape prevention, but it is not al-15 ways possible to interview men who do rape, especially in a college campus context.” The manuscript builds on this assertion in a couple places. I encourage the authors do a little bit more to help readers/the literature on understanding the challenge of eliciting this information.
First: The FGDs are a reasonable way to collect some types of information, but likely not other types of information. The authors go into this in the Limitations (page 11-12, approximately lines 549-560). In a group of 10+ peers from the same community, where their discussions are being recorded and their identities are documented, it seems likely that the information and opinions being shared are preferentially curated by the participants to be more within the perceived norms of the community (or sub-group represented in the room) and to be less representative of any individually-held opinions perceived to be unsupported by the community. The authors should make clear what type of information they believe they collected (which is what is happening lines 549-560) versus what kind of information they believe was likely to have been withheld (does not appear explicitly). Though there are other ways to accomplish this, one way to demarcate what information is/isn’t obtained in this study design is to contrast this design (i.e., FGDs with this kind of membership) with another study design (e.g., interviews with automated prompts; having an individual respondent review the recording and provide commentary). Can the authors propose a complementary study design that might elicit information that this (useful and appropriate) study design is likely to miss?
Second: It probably needs to be more clearly stated in the Limitations that people are not perfect at understanding or explicating their own motivations. The other line of research that our discipline needs to pursue is empirically document how actions-taken are connected to subgroups of perpetrators. For example, a prospective cohort design following college men through their years in HEI and correlating baseline covariates with actions taken later in their time in HEIs, provides us with different types of information that also addresses this problem. Again, I don’t mean to say that the current study design is flawed – but to address the challenge identified in the Abstract, our discipline will need to be clear on the multifaceted, complementary approaches. Your manuscript is quite good and others will learn from it; I encourage you to make it clear to the readers what you folks see as other supporting studies you think should be done here.
I want to call out two sub-sections in the Discussion – Reflexive discussion on the ethics… and Limitations – both are excellent and provide the readers with additional context and insight into how these sessions were held and what was accomplished. Thank you for the details you provided in these sub-sections.
Author Response
Dear Reviewer 1,
Many thanks for the helpful and insightful comments and suggestions on how we can improve our manuscripts.
We have uploaded our responses to your comments.
Best wishes
Yandisa Sikweyiya

Reviewer 2 Report
With such an important topic, what a shame that the authors did not use an appropriate design, especially considering that they themselves note the fatal flaw of the research, which cannot be salvaged: focus group data about a sensitive subject when only individual interviews or anonymous survey data would have been sufficiently valid.
As the authors note: Some participants may have felt obliged to respond in a particular manner in the FGDs, including feeling the need to perform the idealized masculinity and represent themselves positively in the presence of their peers. Also, it is possible that some men sought to represent themselves in a socially desirable manner concerning their gender attitudes and behaviours.
The above isn't just a limitation; it makes the research invalid. Even more concerning, the responses were recorded, possibly making respondents even more self-conscious of their responses.
This point is driven home by the authors' would-be conclusions that, "the study findings reveal how male students in HEIs engage in competition and sexual conquest for women students and discuss this within their peer groups as a way to bond as men while simultaneously dehumanizing women students to rationalize their SV on them".
If it's indeed the case that men, "discuss this within their peer groups as a way to bond as men", that drives home the point that the methodological is irreparably flawed.
Finally, there are way too few citations for the following statement, considering its referred to as a global phenomenon AND its impact on women, children and men!
"Male-perpetrated rape is a global phenomenon that has serious impacts on women, 30 children, and men who experience it (Krug et al. 2002; Jewkes et al. 2006)."
Furthermore, the articles in the lit review should mostly be much more recent than 15+ years ago.
I don't see any way to publish these results given the methodology employed within the context of the topic. If there were quantitative data available as part of the same study that could at least partly address the findings, then it wouldn't be so problematic.
Author Response
Dear Reviewer 2,
Thank you very much for your helpful and insightful comments and suggestions on how we can improve our manuscript.
We have uploaded our responses to your comments.
Best wishes
Yandisa

Round 2
Reviewer 2 Report
I do not believe the data are valid and the authors' response has not changed my assessment of their study.